# PBRM1 acts as a p53 lysine-acetylation reader to suppress renal tumor growth

Weijia Cai [1], Liya Su[1], Lili Liao[1,2], Zongzhi Z. Liu[2], Lauren Langbein [1], Essel Dulaimi[3], Joseph R. Testa [3], Robert G. Uzzo[3], Zhijiu Zhong[4], Wei Jiang[1], Qin Yan [2], Qing Zhang [5,6]* & Haifeng Yang[1]*

p53 acetylation is indispensable for its transcriptional activity and tumor suppressive function. However, the identity of reader protein(s) for p53 acetylation remains elusive. *PBRM1*, the second most highly mutated tumor suppressor gene in kidney cancer, encodes PBRM1. Here, we identify PBRM1 as a reader for p53 acetylation on lysine 382 (K382Ac) through its bromodomain 4 (BD4). Notably, mutations on key residues of BD4 disrupt recognition of p53 K382Ac. The mutation in BD4 also reduces p53 binding to promoters of target genes such as *CDKN1A* (p21). Consequently, the PBRM1 BD4 mutant fails to fully support p53 transcriptional activity and is defective as a tumor suppressor. We also find that expressions of PBRM1 and p21 correlate with each other in human kidney cancer samples. Our findings uncover a tumor suppressive mechanism of PBRM1 in kidney cancer and provide a mechanistic insight into the crosstalk between p53 and SWI/SNF complexes.

[1] Department of Pathology, Anatomy and Cell Biology, Thomas Jefferson University, Philadelphia, PA 19107, USA. [2] Department of Pathology, Yale University, New Haven, CT 06520, USA. [3] Fox Chase Cancer Center, Philadelphia, PA 19111, USA. [4] Sidney Kimmel Cancer Center, Thomas Jefferson University, Philadelphia, PA 19107, USA. [5] Department of Pathology, University of North Carolina at Chapel Hill, Chapel Hill, NC 27599, USA. [6]Present address: Department of Pathology, UT Southwestern Medical Center, Dallas, TX 75390, USA. *email: Qing.Zhang@UTSouthwestern.edu; Haifeng.Yang@jefferson.edu

T*P53*, the tumor suppressive gene (TSG) that encodes p53, plays a pivotal role in suppressing tumor growth mainly through its transcriptional activity[1]. Various post-translational modifications occur on p53, and acetylation on eight key lysine residues was found to be critical for modulation of its transcriptional activity[2]. p53 acetylation can be divided into two groups: acetylation on two lysine residues in p53's DNA-binding domain (K120 and K164), which directly affects its binding to DNA[2–4] and acetylation on six lysine residues in p53's C-terminal domain (CTD) (K370, K372, K373, K381, K382, and K386), which regulates its transcriptional activity through interactions with other proteins, also known as acetylation 'readers'[5–11]. p53 acetylation 'writers' such as histone acetyltransferases and 'erasers' such as histone deacetylases and sirtuins are well described[1], but the identities and functions of p53 acetylation 'readers' remain unclear. CREB-binding protein (CBP) was the only identified CTD acetylation 'reader' which recognizes and binds to p53 acetylated at K382 (K382Ac) and enhances p53 transcription[9]. SET bound to unacetylated CTD and inhibited p53's transcriptional activity[10]. However, CBP and SET fail to fully account for the biological activity of p53's acetylated CTD, suggesting that other reader(s) exists.

In agreement with it being a pivotal TSG, around half of human tumors harbor mutations in *TP53*. Interestingly, clear cell Renal Cell Carcinoma (ccRCC), the most common subtype of kidney cancer, seems to be an exception: only a small subset of tumors harbor mutations in *TP53* (~3%)[12]. This suggests the possibility that p53 tumor suppressor function may be compromised by mutations of other genes in ccRCC tumors. We investigated whether mutations in the Polybromo-1 gene (*PBRM1*) in ccRCC impaired the p53 pathway indirectly. *PBRM1* is mutated in approximately 40% of ccRCC tumors[13]. PBRM1, also called BAF180, functions as a chromatin-targeting subunit of a SWI/SNF chromatin remodeler complex and regulates interferon stimulated gene factor 3[14,15]. Its mutation amplifies the HIF-response and collaborates with *Vhl* mutation to generate ccRCC in mouse models[16–18]. A recent paper reported that the components of SWI/SNF complexes had an overall 20% mutation rate in cancer, which is the second most highly mutated entity next to p53. SWI/SNF mutations are also mutually exclusive with *TP53* mutations in many cancer types[19]. Certain SWI/SNF complex components were reported to interact with p53 and were required for p53 function, especially in senescence[20–25]. Notably, PBRM1 was required for p53-mediated replicative senescence in human primary fibroblasts[26]. Hence, we investigate whether PBRM1, through its six acetyl-lysine binding bromodomains (BDs), functions as a reader of acetylated p53. We find that BD4 of PBRM1 is critical for recognition of K382Ac on p53 and this is critical for PBRM1's tumor suppressor function.

## Results

**PBRM1 and p53 binding is enhanced by DNA damage**. To test whether PBRM1 is a potential acetyl-lysine reader of p53, we first examined whether PBRM1 interacts with p53. The immunoprecipitation results show Flag-tagged PBRM1 bound to endogenous p53 in U2OS osteosarcoma cells (Fig. 1a, left) and human embryonic kidney 293T (HEK293T) cells (Fig. 1a, right). Since p53 is activated after DNA damage, we induced DNA damage in p53-null H1299 lung cancer cells co-transfected with PBRM1 and p53 followed by etoposide or bleomycin treatment and found PBRM1-p53 interactions were enhanced (lane 7 vs. lane 3 or lane 5 on Fig. 1b and lane 9 vs. lane 5 on Fig. 1c). It is important to note that neither the PBRM1 nor the exogenous p53 levels changed upon DNA damage (likely because the high rate of exogenous p53 production exceeds its degradation), suggesting

that the increased interaction may be due to changes in post-translational modifications on these proteins. DNA damage did significantly increase the endogenous p53 protein levels in most cases. In HEK293 cells, p53 pulled down PBRM1, and DNA damage increased the amount of endogenous PBRM1 precipitated by similar amount of endogenous p53 (Fig. 1d). In kidney cancer cell lines ACHN, Caki-1 and Ren-01, increased endogenous p53 associated with more endogenous PBRM1 after DNA damage (Supplementary Fig. 1a–c). To confirm the endogenous interaction is not due to nonspecific binding by antibodies, we performed the immunoprecipitation in HCT116 p53 wild-type colorectal carcinoma cells compared with isogenic p53 null cells. The result showed that p53 antibody immunoprecipitated endogenous PBRM1 only when p53 was present (Fig. 1e).

**Lysine acetylation on the p53 CTD enhances binding to PBRM1**. To address whether the C-terminus of p53 is important to bind PBRM1, we constructed a series of p53 C-terminal truncation mutants (Fig. 2a) and found the affinity of these constructs to PBRM1 was largely diminished by deletion of the p53 CTD (Fig. 2b). Since p53's CTD contains several lysines that are acetylated upon DNA damage and BDs are putative acetylation readers, we reasoned that lysine acetylation may significantly boost the PBRM1-p53 interaction. p53 can be acetylated by acetyltransferases including CBP/p300, TIP60/MOF and PCAF[11]. We overexpressed HA-tagged CBP, p300, TIP60 and PCAF to examine which proteins could enhance the PBRM1-p53 interaction, and only CBP and p300 could do so (Fig. 2c). Considering the role of CBP as a p53 acetylation reader[9], we used p300 to further confirm the enhanced interaction. Since PBRM1 expression was also enhanced by p300, we titrated p300 expression and found p300 indeed significantly enhanced their interaction when PBRM1 and p53 levels were almost equal (Fig. 2d, lanes 5 and 6 vs. lane 4). As expected, p300 greatly increased K382Ac levels on p53 (Fig. 2d), suggesting that it could be a binding signal for PBRM1's BDs. After nicotinamide treatment, which inhibits sirtuins and increases p53 acetylation, the interaction between PBRM1 and p53 was also enhanced (Supplementary Fig. 2a), suggesting that acetylation on p53 might enhance its interaction with PBRM1. K382Ac seems to be naturally occurring without the treatment of DNA damaging agents as it can be detected in the nucleus of cancer cells in human ccRCC tumors (Supplementary Fig. 2b). Moreover, a p53 6KR mutant, in which lysine residues 370, 372, 373, 381, 382 and 386 were mutated to arginine, failed to increase binding to PBRM1 in presence of p300 co-expression, despite similar basal binding to PBRM1 as wild-type p53 (Fig. 2e). This suggests acetylation on the p53 CTD is critical for the enhanced interaction between PBRM1 and p53.

To identify the acetylation site(s) on the p53 CTD responsible for the enhanced PBRM1-p53 interaction, we synthesized biotinylated p53 C-terminal peptides with or without acetylation on key lysines and tested their ability to bind PBRM1. The peptide containing acetylated lysine 382 of p53 displayed significantly higher affinity to endogenous PBRM1 and BRD7, a protein that complexes with PBRM1 in SWI/SNF, in H1299 cells than the non-acetylated control peptide (Fig. 2f, lane 8 vs. lane 3). Similar results were obtained in HEK293T and HCT116 cell lysates (Supplementary Fig. 2b, c), where H3K14Ac is known to be bound by BD2 of PBRM1[27]. We also tested the affinity of these peptides toward exogenous full-length PBRM1 or the PBRM1 BDs (BD1–6). In both cases the p53 peptide with K382Ac showed higher affinity to PBRM1 than the non-acetylated peptide (Fig. 2g, lane 7 vs. lane 2), suggesting that the bromodomians alone were sufficient for the enhanced binding to the p53 K382Ac peptide. Interestingly, the enhanced interaction seemed to be site-specific,

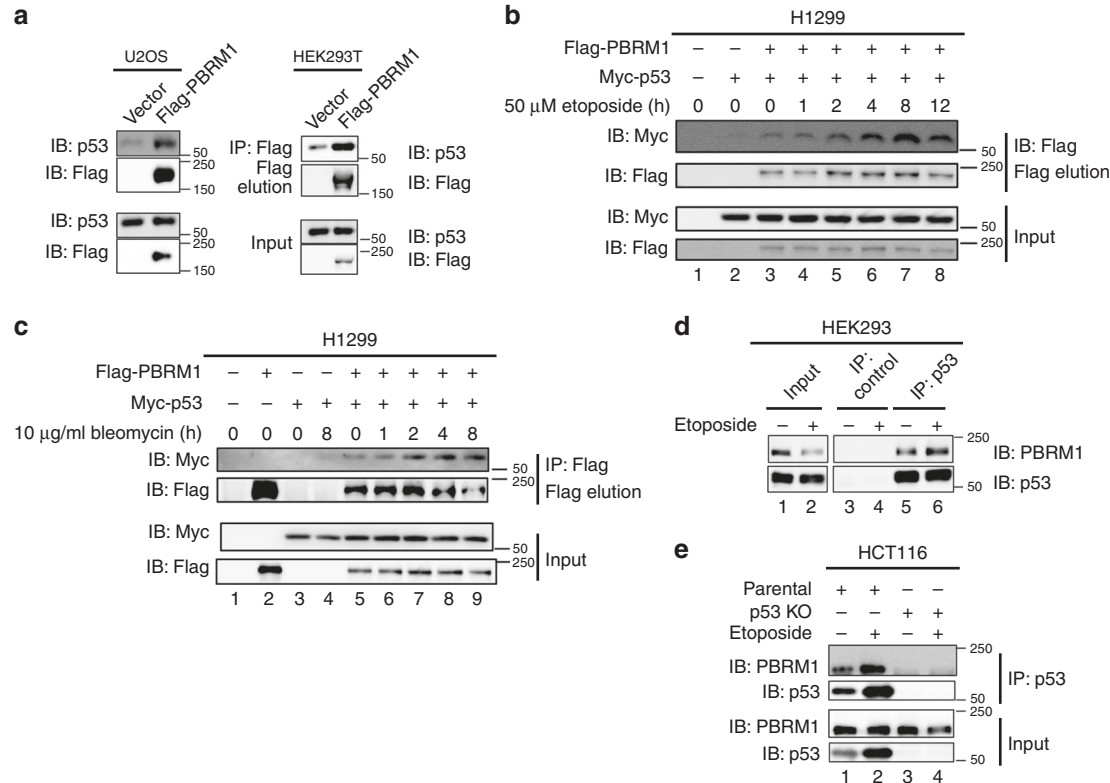

**Fig. 1 PBRM1 and p53 display a physical association that is enhanced after DNA damage. a** U2OS cells (left) and HEK293T cells (right) were transfected with vector or Flag-PBRM1 and harvested for immunoprecipitation with Flag-M2 beads and elution with 3X Flag peptide. Inputs and eluates were analyzed by immunoblots. **b**, **c** H1299 cells were transfected with Flag-PBRM1 and Myc-p53 and treated with etoposide (50 μM, **b**) or bleomycin (10 μg/ml, **c**) for the indicated times. Lysates were subjected to immunoprecipitation with Flag-M2 beads. Inputs and eluates were analyzed by immunoblots. **d** HEK293 cells were treated with vehicle (DMSO) or etoposide (50 μM) for 8 h and harvested for immunoprecipitation with control IgG and p53 antibodies. The bound PBRM1 and p53 were examined by immunoblots. **e** HCT116 and HCT116 p53−/− cells were treated with DMSO or 50 μM etoposide for 24 h. Lysates were immunoprecipitated with p53 antibody. The bound PBRM1 and p53 were examined by immunoblots. Source data are provided as a Source Data file.

as acetylation on other individual sites did not increase the affinity to PBRM1 compared with the non-acetylated peptide. Similarly, in two doubly acetylated peptides K372Ac/K382Ac and K373Ac/K382Ac, only K372Ac/K382Ac showed higher affinity to PBRM1 than the non-acetylated peptide and even the K382Ac peptide (Fig. 2g, lane 9 vs. lanes 7 and 2). Moreover, although the basal bindings were similar, the p53 K382R mutant did not show a p300-dependent increase in binding to PBRM1 as did the wild-type p53 (Fig. 2h). Altogether, these data suggest PBRM1 may recognize particular p53 acetylation pattern/code(s) containing K382Ac.

**BD4 of PBRM1 recognizes p53 K382Ac.** To investigate whether the BDs of PBRM1 interact with p53, we constructed a series of PBRM1 N-terminal truncations (Fig. 3a) and found that deletion up to and including BD4 significantly reduced the interaction between PBRM1 and p53 but not with other SWI/SNF complex components BRG1, BRD7, and BAF57 (Fig. 3b). Moreover, PBRM1's six BDs (BD1–6) without the C-terminal portion lost the ability to bind other SWI/SNF complex components, as previously shown[16], but still bound p53 as well as full-length PBRM1 (Fig. 3b). This suggests that PBRM1 can bind p53 independently of other SWI/SNF components.

On PBRM1, only BD2 has been identified as a reader of histone 3 lysine 14 acetylation (H3K14Ac)[27], and it is not known which BD may be responsible for the recognition of K382Ac on p53. Only BD4 showed detectable affinity to p53 among the six BDs

(Fig. 3c and Supplementary Fig. 3a). BD4 alone bound p53 CTD peptides but did not show selective affinity to the K382Ac peptide, whereas the five other individual BDs failed to bind the peptides (Supplementary Fig. 3b). We postulated that adjacent BD(s) may assist BD4 in recognizing p53 acetylated at K382 since additional BDs assist BD2 in recognizing H3K14Ac, as reported in two recent papers[28,29]. The combination of bromodomains 4 and 5 (BD45) showed enhanced binding to the K382Ac peptide compared to the non-acetylated control (Fig. 3e). To determine which BD is critical for recognition of p53 acetylation, we mutated critical YN residues, previously reported to be required to form a hydrogen bond with the target acetyl-lysine[30,31], to alanines to disrupt bromodomain 4 (BD4*) and 5 (BD5*) (Fig. 3d). The BD4* mutation in BD45 abolished the enhanced binding to the K382Ac peptide whereas the BD5* mutation did not (Fig. 3e). This suggests that BD4, not BD5, is the critical BD of PBRM1 for enhanced binding to K382 acetylated p53. In vitro, a purified GST-BD2345 protein preferentially bound the K382Ac peptide (Supplementary Fig. 3d, e) whereas GST-BD2, -BD4, -BD234 and -BD245 failed to do so (Supplementary Fig. 3c, d). This suggests additional BDs may help facilitate recognition of lysine acetylation by BD4. We identified two BD4 tumor-derived mutations from cBioPortal[32,33] (N601K and E602K, Fig. 3d), and found that they also abolished the increased affinity to the K382Ac peptide (Fig. 3f). In full-length PBRM1, as expected, the BD4* mutation did not affect its affinity to H3K14Ac, a putative substrate of BD2, but specifically abolished its recognition of p53 K382Ac (Fig. 3g). Moreover, the interaction between the

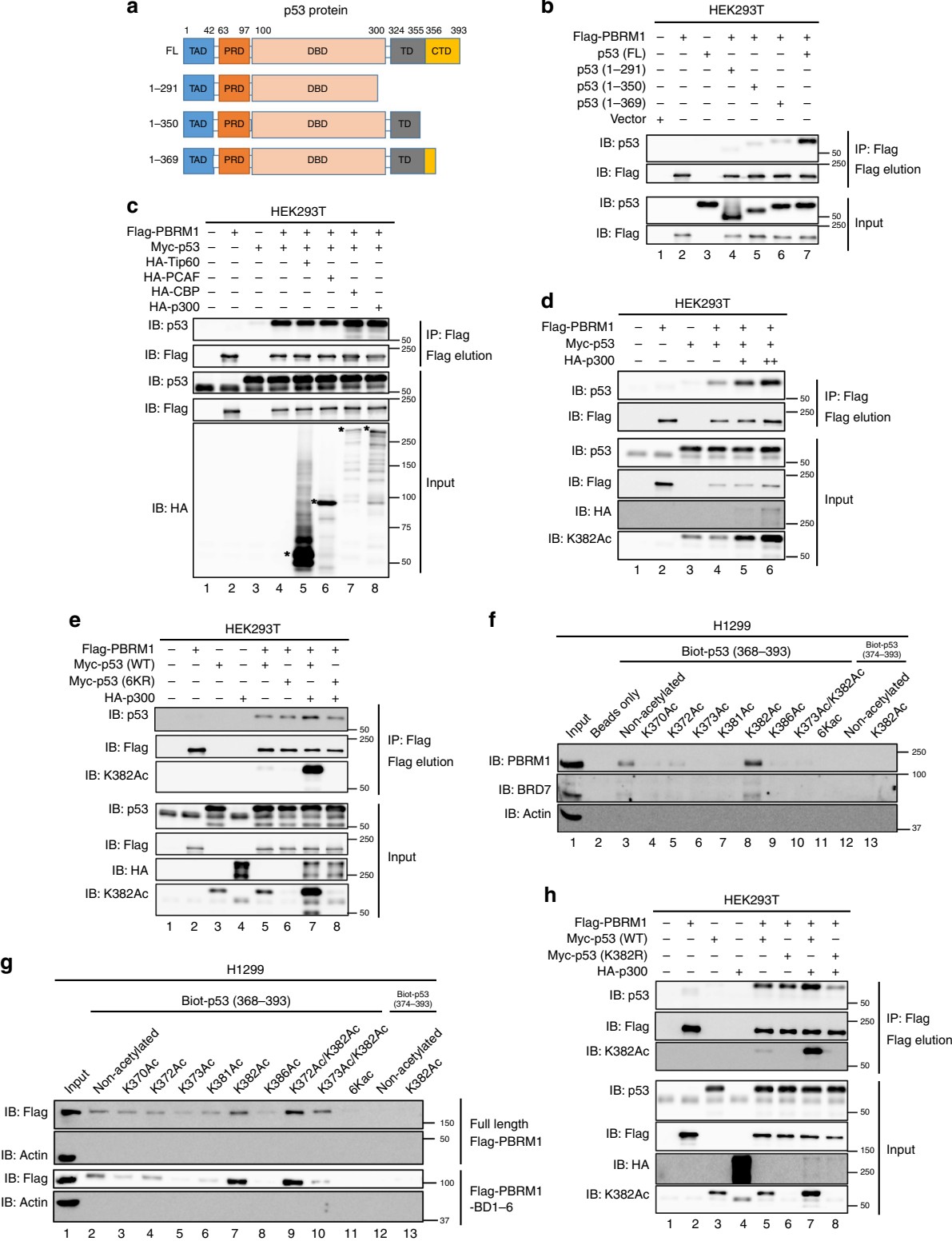

full-length PBRM1 BD4* mutant and p53 was not enhanced by p300 co-expression (Fig. 3h, compare lanes 5–8). These data clearly suggest that BD4 of PBRM1 is critical for recognition of K382Ac on p53.

**PBRM1 loss compromises p53′s transcriptional activity.** To assess PBRM1′s role in p53 function, we knocked out PBRM1 in H1299 cells and found the induction of putative p53 downstream target p21 (encoded by *CDKN1A*) was significantly reduced upon p53 transfection (Fig. 4a, b). A PCR array was performed to examine the expression of most known p53 downstream targets in the parental and PBRM1 knockout (KO) H1299 cells after transfection of p53. The results identified 23 p53 downstream targets that were significantly altered by PBRM1 loss (Supplementary Table 1), and eight of them were further validated by

**Fig. 2 Acetylation on K382 of p53 promotes PBRM1 binding. a** Schematic depiction of the functional domains of p53 and truncated constructs. FL full-length, TAD transactivation domain, PRD proline-rich domain, DBD DNA-binding domain, TD tetramerization domain, CTD C-terminal domain. **b** HEK293T cells were transfected with vectors, Flag-PBRM1, Myc-p53 and Myc-p53 truncated constructs. After immunoprecipitation with Flag-M2 beads, the inputs and eluates were analyzed by immunoblots. **c** Vectors, Flag-PBRM1, Myc-p53 and HA-Tip60, HA-PCAF, HA-CBP or HA-p300 were transfected into HEK293T cells as indicated. Lysates were used for anti-Flag immunoprecipitation and 3X Flag peptide elution. The inputs and eluates were analyzed by immunoblots. **d** HEK293T cells were transfected with Flag-PBRM1, Myc-p53 and HA-p300. The amount of HA-p300 was titrated as indicated. Lysates were used in immunoprecipitations with Flag-M2 beads, and inputs and eluates were analyzed by immunoblots. **e** Vectors, Flag-PBRM1, HA-p300 and Myc-p53 (WT) or Myc-p53 6KR mutant were transfected into HEK293T cells as indicated. Lysates were used for anti-Flag immunoprecipitation and 3X Flag peptide elution. Inputs and eluates were examined by immunoblots. 6KR: lysines mutated to arginines on p53 at K370, K372, K373, K381, K382, and K386. **f, g** Biotinylated p53 peptides with lysine acetylation at the indicated sites were incubated with lysates from H1299 cells (**f**) or H1299 cells transfected with full-length PBRM1 or the PBRM1 bromodomains (Flag-PBRM1 FL or Flag-PBRM1-BD1–6, respectively, **g**). The peptides were pulled down with streptavidin beads and the associated proteins were analyzed by immunoblots. 6KAc: lysines acetylated on p53 at K370, K372, K373, K381, K382, and K386. **h** Vectors, Flag-PBRM1, HA-p300 and Myc-p53 (WT) or Myc-p53 K382R mutant were transfected into HEK293T cells as indicated. Lysates were used for anti-Flag immunoprecipitation and 3X Flag peptide elution. Inputs and eluates were examined by immunoblots. Source data are provided as a Source Data file.

real-time PCR experiments (Fig. 4c). Similar results were obtained in HCT116 PBRM1 knockout cells using etoposide to activate endogenous p53 (Supplementary Fig. 4a, b), whereas the spectrum of altered p53 targets was different from those in H1299 cells. Restoration of PBRM1 expression in two different KO clones excluded the possibilities of off-target and clonal effects, confirming PBRM1 indeed facilitated p53 transcription of select targets (Fig. 4d, e, Supplementary Fig. 4c, d). Moreover, with chromatin immunoprecipitation (ChIP) followed by qPCR, we found p53 expression enhanced the binding of PBRM1 to both the response element 1 (RE1) region and transcriptional start site (TSS) of the *CDKN1A* gene. Consequently, PBRM1 restoration enhanced p53 binding to the RE1 region but not the TSS of *CDKN1A* (Fig. 4f). To examine whether recognition of K382Ac by PBRM1 is critical to PBRM1's role in facilitating p53 transcription, we restored expression of wild-type PBRM1 and the BD4* mutant in H1299 PBRM1 KO cells (Supplementary Fig. 4e). Only wild-type PBRM1 restored the expression of p21 driven by p53 while the BD4* mutant did not (Fig. 4g).

**PBRM1 regulates the p53 signaling pathway in ccRCC cells.** Since PBRM1 is frequently mutated in kidney cancer, we investigated the effects of PBRM1 depletion on the p53 signaling pathway in several kidney cancer cell lines. Knockdown of PBRM1 by short hairpin RNAs (shRNAs) significantly reduced p21 induction, and MDM2 induction in most cases, by p53 upon DNA damage in kidney cancer cell lines ACHN (Fig. 5a and Supplementary Fig. 5a), Caki-1 (Fig. 5b), and Ren-01 (Supplementary Fig. 5b), all of which express both wild-type PBRM1 and p53. Conversely, transient exogenous expression of PBRM1 driven by doxycycline induction (Supplementary Fig. 5c) or transient transfection (Fig. 5c) induced p21 expression in RCC4, a p53-wild type and PBRM1-null kidney cancer cell line. To exclude the possible effects of DNA damage activating other signaling pathways, we used the MDM2 inhibitor Nutlin-3a to selectively accumulate and activate p53. We found that p21 induction by p53 was also reduced by PBRM1 knockdown in ACHN cells (Fig. 5d and Supplementary Fig. 5d). Moreover, over-expressed PBRM1 failed to induce p21 or MDM2 in the absence of p53 in RCC4 cells (Fig. 5c). Our data strongly suggest that PBRM1 is required for full induction of p21 and possibly MDM2 by activated p53 in kidney cancer cells.

In agreement with the results in H1299 PBRM1 KO cells, in RCC4 cells, the PBRM1 BD4* mutant failed to enhance p21 expression after p53 activation, either by DNA damage (Fig. 5e) or Nutlin-3a treatment (Supplementary Fig. 5e), whereas wild-type PBRM1 did. Similarly, the N601K and E602K PBRM1 mutants also failed to enhance p21 expression after DNA damage

(Fig. 5f). Interestingly, PUMA expression was greatly enhanced by wild-type PBRM1 but not the mutants, while MDM2 expression was impacted to a lesser extent (Fig. 5f). We further investigated several p53 downstream target genes in RCC4 cells, and found their expression was up-regulated by restoration of PBRM1 but not PBRM1 BD4* (Fig. 5g). ChIP experiments also showed that p53 binding to the RE1 region on the *CDKN1A* promoter was significantly enhanced by wild-type but not BD4* mutant PBRM1, whereas p53 binding to the TSS region on *CDKN1A* was weak and showed no significant change after PBRM1 expression in RCC4 cells (Fig. 5h). Since p21 is a CDK inhibitor and its activation leads to G1/S cell cycle arrest, we analyzed whether reduced p21 induction after DNA damage in PBRM1-depleted cells affected cell cycle progression in kidney cancer cells. In ACHN cells, PBRM1 knockdown by two different shRNAs significantly decreased the proportion of G1 phase cells and increased the proportion of S phase cells after DNA damage (Fig. 5i, j, and Supplementary Fig. 5i), similar to p53 knockdown or p21 knockdown (Supplementary Fig. 5g–i). This is consistent with each having a defect in G1/S cell cycle arrest and suggests that they share the same pathway to regulate cell cycle progression.

**Regulation of p53 is essential to PBRM1's TSG function.** Next, we sought to determine whether disruption of PBRM1 recognition of acetylated p53 is critical to PBRM1's tumor suppressive function in vivo. We focused on Ren-01 cells which form tumors rapidly in nude mice. We restored wild-type or BD4* mutant PBRM1 expression in Ren-01 PBRM1 KO cells and found the expression of p53 downstream targets was rescued by wild-type but not BD4* mutant PBRM1 after DNA damage (Fig. 6a). To test our hypothesis further, we expressed the same genes in a PBRM1-null ccRCC cell line SLR24 and performed the same experiment described in Fig. 6a. After DNA damage, the expression of p21, PUMA, and MDM2 were significantly induced by wild-type but not BD4* mutant PBRM1 or GFP in SLR24 cells (Fig. 6b). Thus BD4 mutation on PBRM1 abolishes its ability to assist p53 to induce a subset of its downstream targets.

In a nude mouse xenograft model, restoration of wild-type PBRM1 in Ren-01 PBRM1 KO cells suppressed tumor growth when compared with cells expressing either GFP or the BD4* PBRM1 mutant (Fig. 6b, c, e, f). This suggests that PBRM1 inhibits kidney tumor growth and recognition of acetylated p53 is essential to this inhibition. Similar to our in vitro results, p21 expression was also significantly increased in tumors expressing wild-type PBRM1 but not the BD4* mutant (Fig. 6d, g and Supplementary Fig. 6a). To determine the role of the p53 pathway in tumors generated by Ren-01 cells, we knocked down the

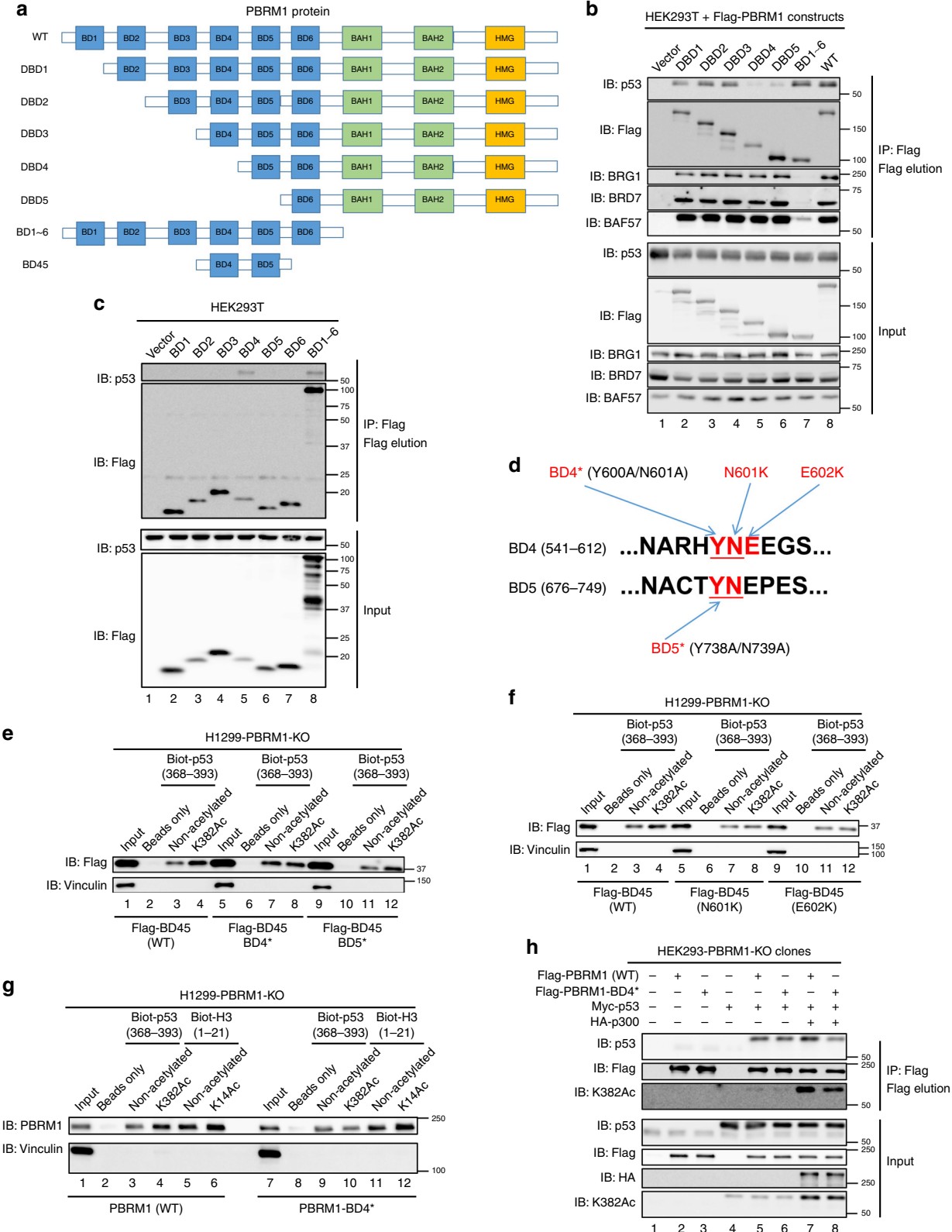

expression of p53 with shRNA (Supplementary Fig. 6b) and examined tumor growth in nude mice. Knockdown of p53 in Ren-01 cells significantly accelerated tumor growth (Supplementary Fig. 6c, d). This suggests that the p53 signaling pathway is tumor suppressive in Ren-01 cells. Taken together, the role of PBRM1 in suppressing tumors generated by Ren-01 cells was likely partially mediated by BD4 recognition of acetylated p53.

**PBRM1 loss is associated with p21 loss in human ccRCC tumors.** To confirm whether PBRM1 indeed regulates p53 function in human kidney tumors, we investigated the relationship between PBRM1 and p21 expression by immunohistochemistry (IHC) in human kidney tumor samples with a ccRCC tissue microarray (Fig. 7a). Across all tumor samples, statistical analysis showed a significant positive correlation

**Fig. 3 Bromodomain 4 of PBRM1 is required for recognition of acetylated K382 on p53. a** Schematic depiction of the functional domains of PBRM1 and truncated constructs. WT: wild-type, BAH: bromo-adjacent homology domain, HMG: high-mobility group domain. **b**, **c** HEK293T cells were transfected with vectors, Flag-WT PBRM1 and Flag-PBRM1 truncated constructs (**b**), and each individual or all the PBRM1 bromodomains (BD1–6, **c**). After immunoprecipitation with Flag-M2 beads, the inputs and eluates were analyzed by immunoblots. **d** Schematic of key amino acid residues of binding pockets in PBRM1 bromodomains 4 and 5. The critical YN residues are underlined and mutations are red. **e–g** H1299 PBRM1 KO cells were transfected with Flag-PBRM1 bromodomains 4 and 5 (Flag-BD45) containing mutations (BD4* or BD5*) that abolish acetyl-lysine recognition in each domain (**e**), Flag-BD45 containing tumor-derived mutations (**f**) or full-length PBRM1 containing the BD4* mutation (**g**). Lysates were incubated with biotinylated p53 peptides with lysine acetylation at the indicated sites. The peptides were pulled down with streptavidin beads and the associated proteins were immunoblotted. **h** Vectors, Myc-p53, HA-p300, and Flag-PBRM1 with or without the BD4* mutation were transfected into HEK293T PBRM1 knockout cells. The lysates were subjected to anti-Flag immunoprecipitation and 3X Flag peptide elution. Inputs and eluates were analyzed in immunoblots. Source data are provided as a Source Data file.

between PBRM1 loss and p21 loss ($p = 0.042$) (Fig. 7b and Supplementary Fig. 7). When the comparison was performed within each grade, grades 1 and 2 showed highly significant positive correlations between these losses ($p = 0.00077$ and $p = 0.0093$, respectively), while grade 3 did not show any and grade 4 showed a weak but significant correlation ($p = 0.044$) (Fig. 7b and Supplementary Fig. 7). When the comparison was performed within each stage, only stage 1 tumors showed a highly significant correlation between PBRM1 and p21 losses ($p = 0.00074$) while the other stages did not display significant correlations (Fig. 7b and Supplementary Fig. 7). This is likely caused by many genetic and epigenetic changes that occur at higher grades or tumor stages that disrupt the correlation between PBRM1 and p21 expression. This analysis suggests that the link between PBRM1 and p21 observed in cell lines is conserved in human tumor samples.

## Discussion

In this report we found that PBRM1 and p53 physically associate. K382 acetylation on p53, which may be modified by p300 and CBP after DNA damage, acts as a binding signal for the BDs of PBRM1. The backbone of p53 peptide might already bind PBRM1, and K382Ac enhances this interaction. We identified BD4 of PBRM1 as the cognate binding partner of K382Ac on p53 (Fig. 7c). Suppression of PBRM1 reduces the induction of many p53 transcriptional targets, and the BD4* PBRM1 mutant that fails to recognize K382Ac on p53 was defective in tumor suppression and regulation of p53 targets (Figs. 4–6). Importantly, the link between PBRM1 and p21 was preserved in human tumor samples (Fig. 7). Thus we hypothesize that DNA damage increases p53 abundance and the K382Ac level, which not only dissociates SET from p53 to relieve the suppression of p53 transcriptional activity, but it also provides a binding signal to PBRM1 for enhanced transcription. PBRM1 binds to both H3K14Ac and K382Ac on p53, and these interactions may retain p53 much longer at its target promoters to promote full transcriptional activation of its targets (Fig. 7c).

p53 has been reported to interact with many SWI/SNF complex components, including SNF5[22], BRG1[23], BAF60a[34], BRD7[20], and ARID1A[21]. Here we report that PBRM1 also physically interacts with p53. It was not previously known whether PBRM1 binds to p53 indirectly via other SWI/SNF subunits. We found that the PBRM1 BDs alone (BD1–6) failed to bind other SWI/SNF complex components yet still retained the ability to bind p53 (Fig. 3b). In agreement with this finding, Flag-BD4 alone bound p53 (Fig. 3c and Supplementary Fig. 3a) and purified GST-BD2345 directly bound the p53 CTD peptide (Supplementary Fig. 3d, e). This shows the interaction between PBRM1 and p53 is direct and not mediated by other SWI/SNF complex components.

Although acetylation is indispensable for p53's transcriptional function and SWI/SNF complexes function as transcriptional co-

factors, no acetylation-dependent interaction has been reported between p53 and SWI/SNF complexes. In this study, we report that acetylation on p53 lysine 382 enhanced the interaction between PBRM1 and p53 via recognition by PBRM1 BD4. A previous report showed that the BDs of PBRM1 did not bind to a p53 K382Ac peptide[27]. The difference is likely due to the length of the peptides used. In our experiments, the p53 peptide used was much longer than theirs (368–393, 26 residues vs. 375–388, 14 residues, respectively) and hence may provide higher affinity due to a greater potential interaction surface. Nevertheless, two shorter peptides lacking K370, K372 and K373 (374–393, 20 residues) failed to interact with PBRM1 (Fig. 2f, lane 12 vs. lane 3 and Fig. 2g, lane 12 vs. lane 2). This suggests that those amino acid residues (368–373), though approximately 10 residues away from the recognition site (K382Ac), were still critical for PBRM1 binding. Since screening analyses between BDs and acetylated substrates have generally used peptides of around 15 residues, they could potentially miss some important discoveries. Two recent papers[28,29] have shown cooperation between BDs of PBRM1 in recognition of H3K14Ac. The same may be true in this instance, that other BD(s) collaborates with BD4 to generate stable recognition of p53 K382Ac (Fig. 3 and Supplementary Fig. 3). Since the structure of the binding between short peptides and single BDs is well analyzed[31,35], the structural insight gained from the enhanced affinity between longer peptides and multiple BDs will be very intriguing. Interestingly, the K382Q mutation on p53 failed to mimic K382Ac to enhance binding to PBRM1 (Supplementary Fig. 2e, f). We conclude that although the K382Q mutation abolishes the positive charge of K382, it failed to fully recapitulate the characteristics of acetylated K382 on p53.

Similar to histone tails, the p53 CTD is enriched in many types of post-translational modifications[36]. Hence the 'histone code' hypothesis[37], in which histone modifications recruit other proteins by specific recognition, likely applies to p53 as well. An interesting finding here was that K372Ac/K382Ac double acetylation significantly enhanced the interaction between PBRM1 and p53 whereas K373Ac/K382Ac did not enhance, and possibly even reduced, the interaction. 6KAc, in which all 6 lysine residues in the p53 CTD were acetylated, diminished the interaction compared to K382Ac (Fig. 2f, g). This suggests that only certain pattern(s) of acetylation can enhance p53 recognition by PBRM1. This hypothesis is logical since recognition by a reader protein should be relatively selective, considering the complexity of post-translational modifications. Notably, we must exercise caution in hypothesizing that all acetylation on the p53 CTD exert similar functions, since peptides with individual acetylation or combined six acetylation exhibit distinct binding to PBRM1. Consistently, a proteomic screen based on p53 CTD without acetylation or with 6KAc identified SET bound to unacetylated p53 CTD but failed to find any protein bound to CTD with 6KAc[10]. This pattern-specific recognition may contribute to fine-tuning p53 transcriptional activity since PBRM1 loss decreased expression of only

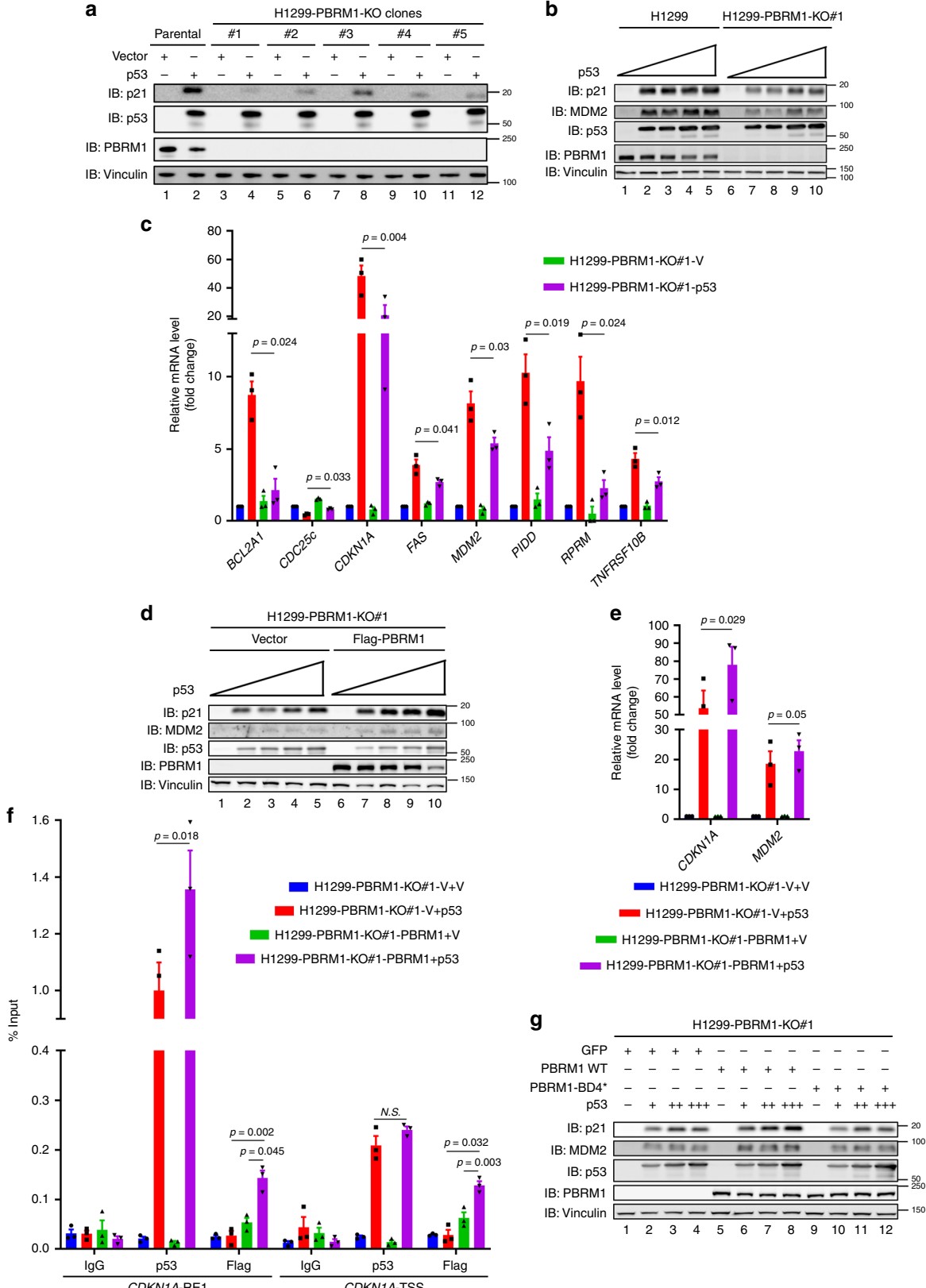

a subset of p53 target genes. In future studies, we will investigate whether different combinations of post-translational modifications, such as acetylation and methylation, interfere with PBRM1′s recognition of K382Ac.

The transcriptional activity of p53 is compromised by PBRM1 loss. Mechanistically, we found p53 binding to the *CDKN1A* promoter (RE1 region) was enhanced by PBRM1 (Fig. 4f and Fig. 5h). p53 activation also recruited PBRM1 binding to both the promoter and TSS of *CDKN1A* (Figs. 4f and 5h). Unexpectedly, the PBRM1 BD4* mutant, which failed to recognize p53 K382Ac but recognized H3K14Ac (Fig. 3g), bound the *CDKN1A* gene as well as wild-type PBRM1 (Fig. 5h). Moreover, there was

**Fig. 4 PBRM1 is required for full p53 transcriptional activity on a subset of its targets. a** p53 was transfected into H1299 parental cells and five PBRM1 knockout clones, and lysates were immunoblotted. **b** H1299 parental and PBRM1-KO#1 cells were transfected with increasing amounts of p53 and lysates were immunoblotted. **c** H1299 parental and PBRM1-KO#1 were transfected with vector or p53, and RNA was extracted and reverse-transcribed. After PCR array, significantly altered p53 target genes were further validated by qPCR. The expression of significantly altered genes is shown as mean ± SEM. p-values were calculated using the paired two-tailed Student's t-test. **d–f** Increasing amounts of p53 were transfected into H1299 PBRM1-KO#1 with or without Flag-PBRM1 re-expression and lysates were analyzed by immunoblots. Cells with comparable p53 expression were harvested for RT-qPCR (**e**) or ChIP followed by qPCR (**f**). Data are shown as mean ± SEM. **g** H1299 PBRM1-KO#1 cells stably expressing GFP, wild-type or BD4* mutant PBRM1 were transfected with increasing amounts of p53. Lysates were analyzed by immunoblots. Source data are provided as a Source Data file.

comparable PBRM1 binding to the RE1 and TSS whereas p53 binding to RE1 was significantly higher than to the TSS (Figs. 4f and 5h). This suggests PBRM1 was not recruited to the *CDKN1A* gene via p53. p53 has been reported to recruit acetyltransferases such as p300 to acetylate histones and facilitate transcription[38]. Now, it seems likely that p53 may trigger histone acetylation, which recruits PBRM1 via recognition by its BDs. Nevertheless, the acetylation-dependent interaction between PBRM1 and p53 is critical for increased p53 binding at the *CDKN1A* promoter since the BD4* mutant failed to enhance p53 binding to RE1 (Figs. 5h and 7c).

Our in vitro and in vivo results, including kidney tumor patient samples, showed loss of PBRM1 in ccRCC dampened p53 function and especially p21 expression, which is key for cell cycle arrest and senescence. This is also consistent with previous reports showing that PBRM1 knockout in mouse kidneys permits continued cell proliferation through inhibition of replication stress[39]. Replication stress is frequently bypassed by altered p53 pathway in other types of cancers. Lee et al. found PBRM1 deficiency induced p21 expression and consequent senescence in mouse hematopoietic stem cells[40]. This may be attributed to the tissue-specificity of PBRM1 function and may explain why most PBRM1 mutations are found in renal cancer. Gao et al. found that knockdown of PBRM1 reduced p21 levels in approximately half of their tested shRNAs, but they did not draw a clear conclusion[16]. In our experiments, several techniques including knockdown, knockout, overexpression and restored expression of PBRM1 confirmed that PBRM1 regulates p21 expression, and this effect is much more pronounced after p53 activation. The PBRM1 BD4* mutant, which is highly specific in its inability to support the p53 pathway but being able to recognize H3K14Ac, compromised the tumor suppressive function of PBRM1 (Fig. 6e, f). We recently discovered that ISGF3 regulation by PBRM1 is one of its important tumor suppressor functions in ccRCC[15], and our evidence here suggest that supporting p53 pathway is another important tumor suppressor function of PBRM1. p21 in combination with other p53 targets may channel this tumor suppressor function. In ccRCC, p53 mutations are rare probably because frequent PBRM1 mutations already partially disable p53 function, reducing the selection pressure for mutant p53. Since PBRM1 deficiency only compromise, not abolish, p53 function on a subset of p53 targets, it is unlikely to recapitulate all the attributes of p53 deficiency such as high grade and aggressiveness. However, a partially disabled p53 pathway may still be important for ccRCC tumorigenesis and/or tumor growth, and this could have therapeutic implications for modulating the p53 pathway to treat ccRCC patients. In summary, our findings identified PBRM1 as a functional p53 acetylation reader, elucidated the role of the PBRM1-p53 axis on renal tumor growth, and provided insights on the crosstalk between SWI/SNF complexes and the p53 pathway.

## Methods

### Chemicals, reagents, and antibodies.
Etoposide (S1225), bleomycin (9041–93–4) and trichostatin A (TSA, S1045) were purchased from Selleck Chemical. Nutlin-3a (18585) was purchased from Cayman Chemical. Propidium iodide (PI, P3566) was purchased from Life Technologies. RNaseA (EN0531), Pierce Protease Inhibitor Tablets (A32963) and Pierce Phosphatase Inhibitor Mini Tablets (A32957) were purchased from Thermo Scientific. Puromycin (P-7255), G418 (A1720), Doxycycline (D9891), and Nicotinamide (NAM, 72340) were purchased from Millipore Sigma.

Antibodies for vinculin (sc-73614), actin (sc-8432), p53 (FL-393, sc-6243 and DO-1, sc-126) and p21 (sc-6246) were purchased from Santa Cruz Biotechnology. Antibodies for GST (#2625), p21 (#2947), PUMA (#12450) and Myc (#2040) were purchased from Cell Signaling Technology. Anti-PBRM1 (A301–591A), -BRD7 (A302–304A) and -BAF57 antibodies (A300–810A) were purchased from Bethyl Laboratories. Antibodies for Flag (M2, F3165), MDM2 (OP145) and γH2Ax (05–636) were purchased from Millipore Sigma. Anti-HA antibody (901514) was purchased from Biolegend. Anti-p53 K382Ac antibody (GTX62061) was purchased from GeneTex. For western blotting, except for vinculin (sc-73614) which was used 1:2000 dilution, all other Santa Cruz Biotech antibodies were used at 1:200 dilution. All the other antibodies from different companies were used at 1:1,000 dilution for western blot, except for anti-Flag antibody was used at 0.5 μg/ml. All the original blots used for figures were included in the PBRM1-p53-Source Data file.

### Plasmids.
The Flag-PBRM1 plasmid was constructed by cloning PBRM1 cDNA into p3XFlag-CMV10 (Sigma-Aldrich). The Myc-p53 plasmid was constructed by cloning p53 cDNA into pcDNA3-Myc. The GST-PBRM1 constructs were generated by subcloning the BDs into the pGST parallel expression vector[41].

pCI-Flag-PCAF originated from Nakatani lab (Addgene plasmid #8941)[42]. pCMVβ-p300-Myc and pcDNA3β-Flag-CBP-HA were from Tso-Pang Yao's alb (Addgene plasmid #32908)[43]. HA-PCAF, -CBP, and -p300 were generated by subcloning into a pcDNA3-HA vector. HA-Tip60 plasmid was constructed by cloning the cDNA into a pcDNA3-HA vector.

Lentiviral Flag-PBRM1 and mutants were generated by subcloning PBRM1 into a pLNCX-GFP lentiviral vector (a gift from Dr. Wei Xu)[44]. Mutations in PBRM1 and p53 were generated using the QuikChange Site-Directed Mutagenesis Kit (Stratagene) according to the manufacturer's instructions.

Doxycycline-inducible PBRM1 was constructed by subcloning PBRM1 into the pTetOne vector (Clontech). pLKO.1 lentiviral shRNAs against PBRM1 (#1: TRCN0000015994, #2: TRCN0000235890), p53 (TRCN0000003755) and p21(#1: TRCN0000287021, #2: TRCN0000294421) were purchased from Sigma-Aldrich. pLKO.1 scramble (SCR) shRNA was from David Sabatini's lab (Addgene plasmid #1864)[45].

For p53 knockout, p53 Double Nickase Plasmid (h) (sc-416469-NIC) was purchased from Santa Cruz Biotechnology. For PBRM1 knockout, sgRNA primers (PBRM1-sg2A-F: CACCGTCATCCTTATAGTCTCGGA, PBRM1-sg2A-R: AAACTCCGAGACTATAAGGATGAC, PBRM1-sg2B-F: CACCGCTCTGTGAGCTCTTCATTA, PBRM1-sg2B-R: AAACTAATGAAGAGCTCACAGAGC, PBRM1-sg7A-F: CACCGGCGAGGAGATCTATATCTT, PBRM1-sg7A-R: AAACAAGATATAGATCTCCTCGCC, PBRM1-sg7B-F: CACCGCCAAAACTTATAATGAGCC, PBRM1-sg7B-R: AAACGGCTCATTATAAGTTTTGGC, PBRM1-sg20A-F: CACCGTGGCAACCTGGTTCACCAT, PBRM1-sg20A-R: AAACATGGTGAACCAGGTTGCCAC, PBRM1-sg20B-F: CACCGGCTCCATTACAATGACATG and PBRM1-sg20B-R: AAACCATGTCATTGTAATGGAGCC) were designed using Optimized Crispr Design (crispr.mit.edu) and cloned into pX335-U6-Chimeric_BB-CBh-hSpCas9n(D10A)[46]. All constructs were confirmed by sequencing.

### Cell culture and generation of stable cell lines.
HEK293, HEK293T, ACHN, RCC4, Ren-01, SLR24, and NCI-H1299 (H1299) cells were maintained in DMEM supplemented with 10% FBS. U-2 OS (U2OS), Caki-1, HCT116, and HCT116 p53−/− cells were maintained in McCoy's 5A supplemented with 10% FBS. All cells were maintained in incubators with 5% CO$_2$ at 37 °C. HEK293, Caki-1, and ACHN cells were obtained from ATCC. SLR24 cell line was obtained from Dr. William Kaelin's lab at Dana-Farber Cancer Institute.

To establish stable clones with knockdown or restoration of specific proteins, cell lines were infected with lentivirus containing the indicated pLKO.1 shRNA constructs or pLNCX constructs followed by selection with puromycin (2 μg/ml) for 1 week or G418 (1 mg/ml) for 2 weeks, respectively.

To establish clones with knockout of PBRM1 or p53 or inducible expression of PBRM1, cell lines were transfected with pX335 constructs plus a linear puromycin

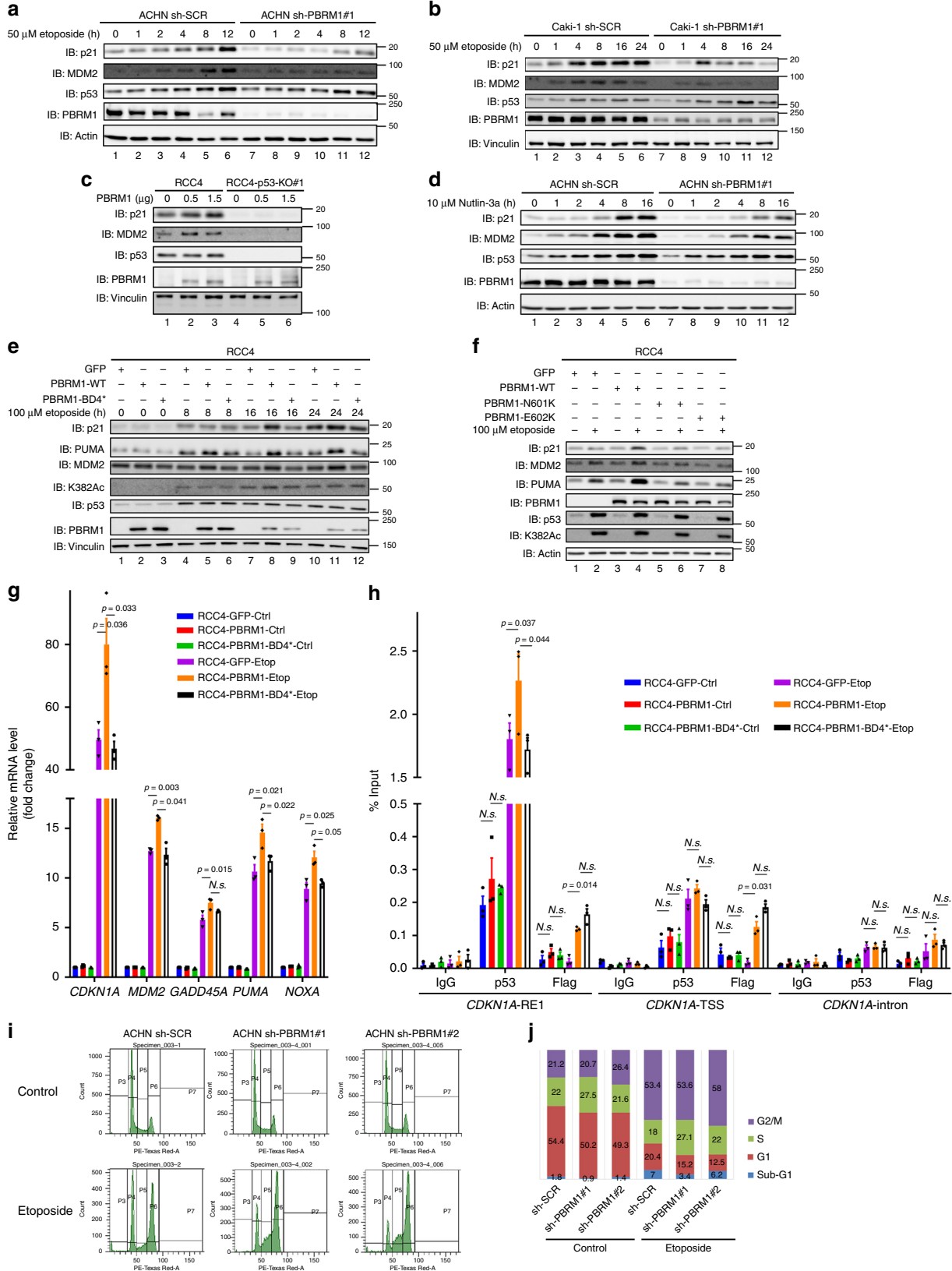

marker (631626, Clontech), p53 Double Nickase Plasmid (h) (sc-416469-NIC, Santa Cruz Biotechnology), or pTetOne construct plus linear puromycin marker using Lipofectamine™ 2000 (11668019, Thermo Scientific) or X-tremeGENE™ HP DNA Transfection Reagent (6366546001, Sigma-Aldrich). After 24 h, cells were selected with puromycin (2 μg/ml) for 3 days. Surviving cells were diluted and transferred into 15 cm dishes. After 7–10 days, single colonies were isolated. After 1–2 weeks, cells were harvested for immunoblot analysis to examine the expression

of specific proteins. For inducible PBRM1 expression, RCC4 cells were treated with 1 μg/ml doxycycline for the indicated times prior to harvest.

**Immunoblot and immunoprecipitation**. After washing with ice-cold PBS, cells were scraped on ice and centrifuged at $10,000 \times g$ for 1 min at 4 °C. For immunoblot analysis, cell pellets were lysed in 1% SDS containing a protease inhibitor cocktail

**Fig. 5 K382Ac recognition by PBRM1 is critical for p53 function in ccRCC cells. a, b** ACHN (**a**) or Caki-1 (**b**) control and PBRM1 shRNA knockdown cells were treated with 50 µM etoposide for the indicated time points. Lysates were analyzed by immunoblots. **c** RCC4 cells with or without p53 KO were transfected with vector or Flag-PBRM1 at the indicated amounts. Lysates were analyzed by immunoblots. **d** ACHN control and PBRM1 shRNA #1 knockdown cells were treated with 10 µM Nutlin-3a for the indicated time points. Lysates were analyzed by immunoblots. **e** RCC4 cells stably expressing GFP, wild-type PBRM1, or BD4* mutant PBRM1 were treated with 100 µM etoposide for the indicated time points. Lysates were analyzed by immunoblots. **f** RCC4 cells stably expressing GFP, wild-type or indicated PBRM1 mutants were treated with 100 µM etoposide for the indicated times, and lysates were examined via immunoblots. **g, h** RCC4 cells stably expressing GFP, wild-type or BD4* mutant PBRM1 were treated with 100 µM etoposide for 24 h. Cells were harvested for RT-qPCR (**g**) or ChIP followed by qPCR (**h**). Data are shown as mean ± SEM. p-values were calculated using the paired two-tailed Student's t-test. **i, j** ACHN control and PBRM1 shRNA knockdown cells were treated with vehicle or 50 µM etoposide for cell cycle analysis. Representative data is shown in **i**. Quantification of cell cycle phase is shown in **j**. Source data are provided as a Source Data file.

followed by sonication with a FB50 sonic dismembrator (Fisher Scientific) at 20% amplitude for 15 s twice. After centrifuging for 10 min at 14,000 × g, the supernatants were combined with 5X SDS loading buffer (5% mercaptoethanol, 0.05% Bromophenol blue, 30% glycerol, 10% SDS, 250 mM Tris-HCl (pH 6.8), boiled for 5 min and used in SDS-PAGE. After standard western blot procedures, the blots were developed using the ImageQuant LAS 4000 (GE healthcare Life Sciences) or ChemiDoc MP Imaging System (Bio-rad) with HyGLO™ Chemiluminescent HRP Detection Reagent (E2500, Denville Scientific) or Immobilon™ Western Chemiluminescent HRP Substrate (WBKLS0500, Millipore).

For anti-Flag immunoprecipitation, cell pellets harvested from one 10 cm culture dish were resuspended in 500 µl hypotonic buffer (10 mM Tris-HCl, pH 8.0, 1.5 mM MgCl$_2$, 10 mM KCl) with protease and phosphatase inhibitors, 2 µM TSA and 10 mM NAM, and samples were incubated on ice for 15 min. Twenty-five microliters of Triton X-100 was added and samples were vortexed for 10 s. After centrifuging for 30 s at 14,000 × g, supernatants were removed and pellets were resuspended in 100 µl hypertonic buffer (20 mM Tris-HCl pH 8.0, 420 mM KCl, 1.5 mM MgCl$_2$, 20% (v/v) glycerol) with protease and phosphatase inhibitors, 2 µM TSA and 10 mM NAM and vortexed for 10 s. Pellets were incubated on ice for 40 min and vortexed for 15 s every 10 min. After centrifuging for 10 min at 14,000 × g, supernatants were collected and diluted with 300 µl EBC buffer (50 mM Tris-HCl (pH 7.5), 120 mM NaCl, 0.5% NP-40) with protease and phosphatase inhibitors, 2 µM TSA and 10 mM NAM. After centrifuging for 10 min at 14,000 × g, supernatants were incubated with anti-Flag M2 affinity gel (A2220, Sigma-Aldrich), for 2 h at 4 °C. Beads were washed four times with EBC buffer and eluted with 3X Flag peptide (F4799, Sigma-Aldrich) overnight at 4 °C. The input and Flag-elution were analyzed by SDS-PAGE and immunoblot.

For endogenous p53 immunoprecipitations, cell pellets were lysed in EBC buffer with protease and phosphatase inhibitors, 2 µM TSA and 10 mM NAM. Lysates (approximately 2 mg of total protein) were incubated with 2 µg p53 antibody (DO-1) overnight at 4 °C followed by incubation with protein A/G plus Sepharose (sc-2003, Santa Cruz Biotechnology) for 2 h. Beads were washed with EBC buffer four times, mixed with 1X SDS loading buffer (1% mercaptoethanol, 0.01% Bromophenol blue, 6% Glycerol, 2% SDS, 50 mM Tris-Cl (pH 6.8)) and analyzed by SDS-PAGE and immunoblot.

**Peptide pull-down assay.** Biotinylated peptides with or without acetylated lysine were synthesized by Anaspec (Fremont, California). The peptides contained residues 1–21 of histone H3 (ARTKQTARKSTGGKAPRKQLT), residues 368–393 of p53 (HLKSKKGQSTSRHKKLMFKTEGPDSD) or residues 374–393 of p53 (GQSTSRHKKLMFKTEGPDSD). Cells were lysed in EBC buffer with protease and phosphatase inhibitors, 2 µM TSA and 10 mM NAM. Lysates were incubated with peptides for 1 h at 4 °C. Streptavidin beads (20347, Thermo Scientific) were added for further 1 h incubation. After washing four times with EBC buffer, the bound proteins were boiled with 1X SDS loading buffer and examined by immunoblots.

**Recombinant protein purification.** pGST parallel expression plasmids with PBRM1 BDs were individually expressed in E. coli strain BL21. The expression of the protein was induced by 0.1 mM isopropyl-β-d-thiogalactoside (IPTG, I6758, Millipore Sigma) for 20 h at 18 °C. The GST-tagged proteins were purified with glutathione-Sepharose 4B (GE Healthcare, Catalog #17–5130–01) according to the manufacturer's protocol and eluted with 10 mM reduced glutathione (G4251, Millipore Sigma). To remove glutathione and concentrate proteins, the eluates were spun using Amicon Ultra-15 Centrifugal Filter Unit (UFC901096, Millipore Sigma) with dialysis buffer (25 mM Tris-HCl (pH 8.0), 100 mM NaCl).

**Real-time RT–PCR analyses and p53 PCR array.** Total RNA was extracted from cells using the RNeasy Plus Mini Kit (74136, Qiagen), and the concentration was measured using the NanoDrop ND-1000 system (Thermo Scientific). Reverse transcription was performed using the First Strand cDNA Synthesis Kit (NP100042, Origen) with 0.5 µg total RNA. Real-time PCR was performed with Maxima SYBR Green qPCR Master Mix (K0253, Thermo Scientific) using the Roche LightCycler 480. The primers used for real-time PCR are listed as follows: q-GAPDH-F: GGAGCGAGATCCCTCCAAAAT, q-GAPDH-R: GGCTGTTGTCA TACTTCTCATGG, q-CDKN1A-F: TACCCTTGTGCCTCGCTCAG, q-CDKN 1A-R: GAGAAGATCAGCCGGGCGTTT, q-MDM2-F: GAATCTACAGGGACGC CATC, q-MDM2-R: TCCTGATCCAACCAATCACC, q-BAX-F: CCGCCGTGGA CACAGAC, q-BAX-R: CAGAAAAACATGTCAGCTGCCA, q-PUMA-F: GGGCC CAGACTGTGAATCCT, q-PUMA-R: ACTTGCTCTCTCTAAACCTAT, q-NOXA-F: GTGTGCTACTCAACTCAG, q-NOXA-R: ATTCCTCTCAATTA-CAATGC, q-GADD45a-F: GAGAGCAGAAGACCGAAAGGA, q-GADD45a-R: CAGTGATCGTGCGCTGACT, q-APAF1-F: AAAAGGGGATAGAACCAGAGG, q-APAF1-R: TGCGGCACCTCAAGTCTTC, q-BAI1-F: GCAAACCAAGTTCT GCAACAT, q-BAI1-R: CTCCAGCTCGACCACTCATT, q-BCL2A1-F: AGTGC-TACAAAATGTTGCGTTC, q-BCL2A1-R: gGCAATTTGCTGTCGTAGAAGTT, q-BTG2-F: GCGAGCAGAGGCTTAAGGT, q-BTG2-R: GGGAAACCAGTGGT GTTTGTA, q-CDC25c-F: TCCCTGAAAGATCAAGAAGC, q-CDC25c-R: CC TTGGAAAAATCACCAATC, q-FAS-F: GGGGTGGCTTTGTCTTCTTCTTTTG, q-FAS-R: ACCTTGGTTTTCCTTTCTGTGCTTTCT, q-PIDD-F: TCTGACAC GGTGGAGATGTTCG, q-PIDD-R: AGGTGCGAGTAGAAGACAAAGCAG, q-RPRM-F: AGCAAACCTGTCGGAGTCAA, q-RPRM-R: CTCCCCGCATTC-CAAGTAAG, q-TNFRSF10B-F: CTCTGAGACAGTGCTTCGATGACT and q-TNFRSF10B-R: CCATGAGGCCCAACTTCCT. GAPDH was used as an internal control. Data are shown as mean ± SEM for three independent experiments.

The p53 Signaling Pathway RT$^2$ Profiler PCR Array (PAHS-027Z, Qiagen), consisting of 84 genes related to p53-mediated signal transduction, was used to profile H1299-PBRM1-KO#1 cells transfected with p53 according to the manufacturer's instructions. Briefly, H1299 and H1299-PBRM1-KO#1 cells were transfected with vector or p53 for 24 h before RNA was extracted. The expression of p53 was comparable by immunoblot detection. RNA was reverse transcribed using the RT$^2$ First Strand Kit (330404, Qiagen). The cDNA was mixed with RT$^2$ SYBR Green qPCR Master Mix (330502, Qiagen), and equal aliquots of this mixture (25 µl) were added to each well of the same PCR Array plate that contained the predispensed gene-specific primer sets. Real-time PCR and data collection were performed on the LightCycler 480 (Roche). Data were analyzed on http://www.sabiosciences.com/pcrarraydataanalysis.php.

**Chromatin immunoprecipitation (ChIP).** ChIP assays were performed with the Pierce Agarose ChIP Kit (26156, Thermo Scientific) according to the manufacturer's instructions with minor revisions. In brief, samples were crosslinked with 1% paraformaldehyde and halted by 125 mM glycine. After washing twice with ice-cold PBS, cells were harvested and suspended in Lysis Buffer 1 and incubated on ice for 10 min. 1 × 10$^7$ nuclei were resuspended in 500 µl MNase Digestion Buffer containing 6.25 U Micrococcal Nuclease and incubated at 37 °C for 15 min. After the reaction was stopped, nuclei were pelleted by centrifugation and resuspended in 250 µl Lysis Buffer 2. Nuclei were sonicated at 10% amplitude for 10 s twice and incubated on ice for 30 min. After centrifugation, 50 µl supernatant containing the digested chromatin from 2 × 10$^6$ nuclei was diluted in 1× IP Dilution Buffer and incubated with antibodies (normal rabbit IgG for control, 1 µg FL393 + 1 µg DO-1 for p53 or 2 µg Flag M2 for Flag-PBRM1) overnight at 4 °C. Twenty liters of ChIP Grade Protein A/G Plus Agarose was added and incubated for 1 h at 4 °C. The agarose resins were washed once with IP Wash Buffer 1, twice with IP Wash Buffer 2 and once with IP Wash Buffer 3. 150 µl 1× IP Elution Buffer was added to the washed resin and incubated at 65 °C for 30 min with shaking. The eluate was treated with Proteinase K and DNA was extracted with columns provided in the kit. The resulting purified DNA was used in real-time PCR detection with the primers listed as follows: qCHIP-p21-RE1-F: AGCAGGCTGTGGCTCTGATT, qCHIP-p21-RE1-R: CAAAATAGCCACCAGCCTCTTCT, qCHIP-p21-TSS-F: TATATCAGGGCCGCGCTG, qCHIP-p21-TSS-R: GGCTCCACAAGGAACTG ACTTC, qCHIP-p21-intron1-F: AGTCACTCAGCCCTGGAGTCAA and qCHIP-p21-intron1-R: GGAGAGTGAGTTTGCCCATGA.

**Animal and mouse xenograft tumorigenesis assay.** All animal experiments were performed following the Guide for the Care and Use of Laboratory Animals of the National Institutes of Health and protocol 01462–935A approved by the Thomas

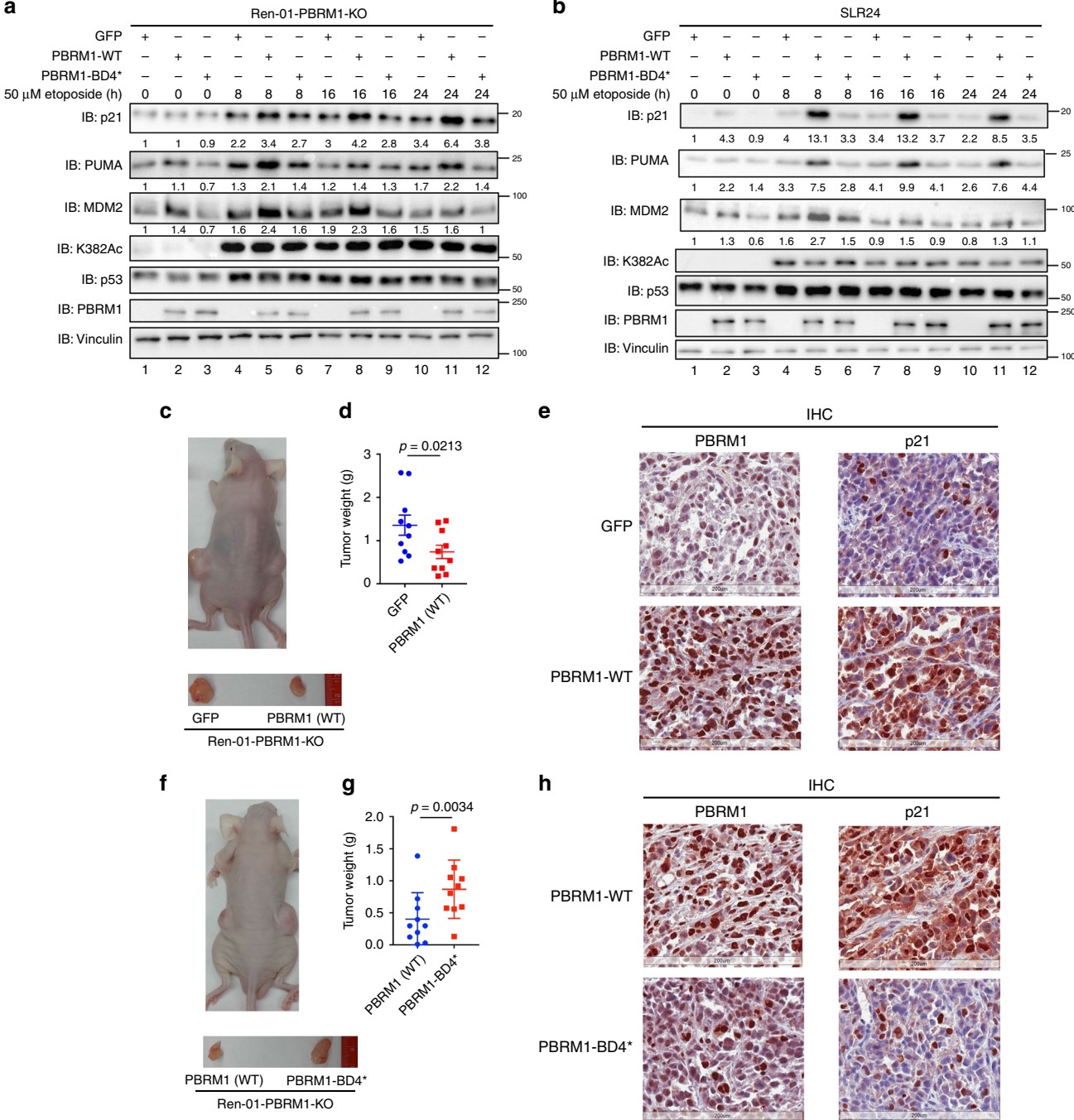

**Fig. 6 Mutation of BD4 abolishes the tumor suppressive function of PBRM1. a, b** GFP, wild-type or BD4* mutant PBRM1 were stably expressed in Ren-01 PBRM1 KO cells (combination of three clones) (**a**) or PBRM1-null SLR24 cells (**b**). Cells were treated with 50 μM etoposide for the indicated times, and lysates were analyzed via immunoblots with indicated antibodies. The band intensity of indicated protein was measured with Bio-rad Image Lab 4.1, and the relative ratios were calculated over the signal intensity of Vinculin in the corresponding lanes. **c–e**. Nude mice xenograft analyses were performed with Ren-01 PBRM1 KO cells expressing GFP (left flank) or PBRM1 (right flank). Representative photographs of a mouse (**c**, top) and tumors (**c**, bottom). Tumors were excised and weighed, and data are presented as mean ± SEM (**d**). p-values were calculated using the paired two-tailed Student's *t*-test. Tumors were stained for PBRM1 and p21 expression via IHC (**e**). Scale bar: 200 μm. **f–h** Nude mice xenograft analyses were performed with Ren-01 PBRM1 KO cells expressing wild-type PBRM1 (left flank) or the BD4* mutant PBRM1 (right flank). Tumors were excised and weighed, and data are presented as mean ± SEM (**g**). *p*-values were calculated using the paired two-tailed Student's *t*-test. Representative photographs of a mouse (**f**, top) and tumors (**f**, bottom). Tumors were stained for PBRM1 and p21 expression via IHC analysis (**h**). Scale bar: 200 μm. Source data are provided as a Source Data file.

Jefferson University Animal Care and Use Committee. $1 \times 10^7$ cells in 100 μl PBS were injected into one side of the dorsal flank of 4–6-week-old male Nu/J nude mice (Charles River). The same number of cells of another cell line were injected into the other flank of the same mouse. Ten mice were analyzed in this manner per experiment. Tumor growth was monitored regularly for up to 14 days. The mice

were sacrificed and tumors were dissected and weighed. Significance differences in tumor weight were determined by the two-tailed Student's paired *t*-test.

**Tissue microarray (TMA) and IHC**. 160 patients diagnosed with ccRCC were selected from Fox Chase cancer Center with the protocol approved by Fox Chase

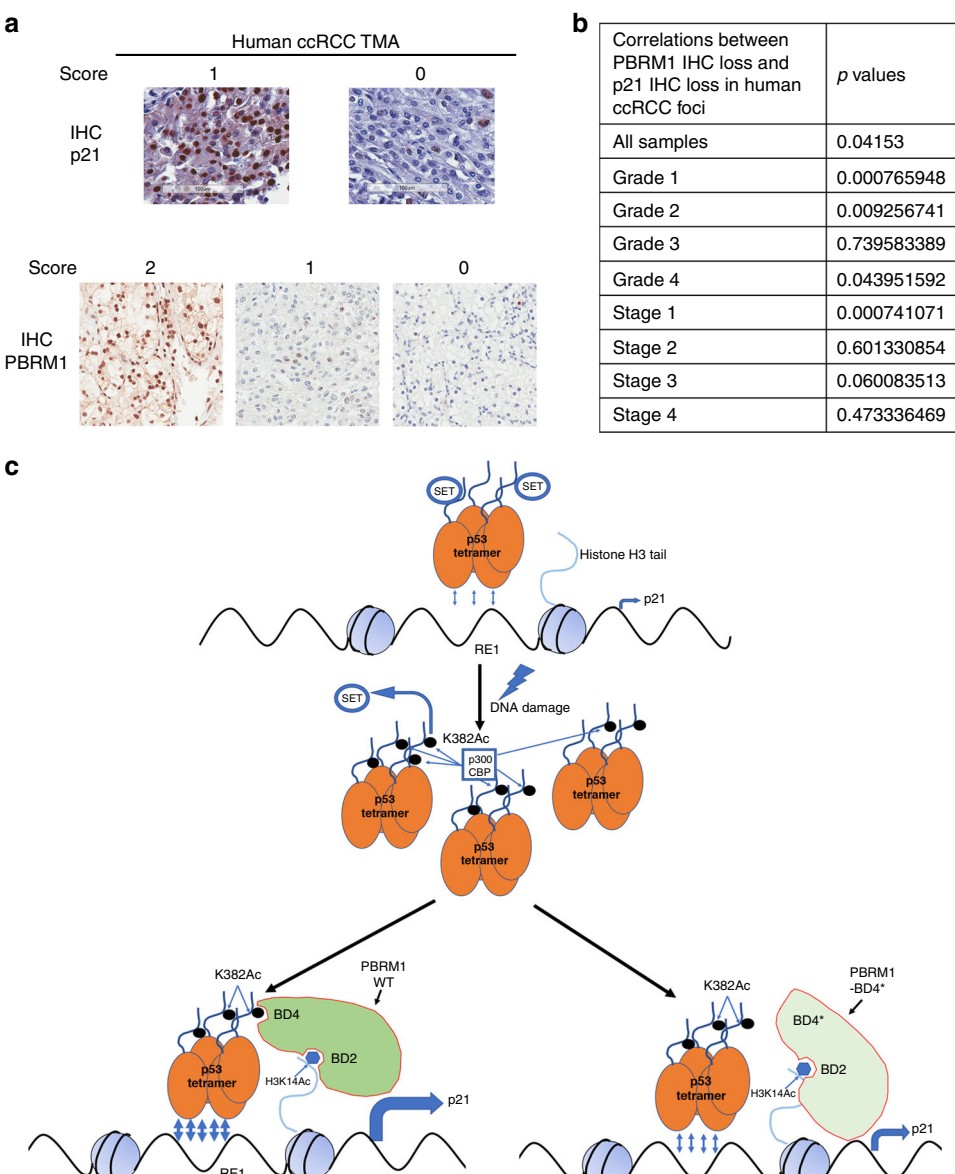

**Fig. 7 PBRM1 loss and p21 loss positively correlate in human ccRCC tumors. a** 160 ccRCC tumors (40 tumors per stage) were used to generate a tissue microarray (TMA). Four foci from different regions of each tumor were selected. PBRM1 expression was analyzed and scored in our previous report[47,48] (**a**, bottom). p21 expression was analyzed and scored in the tumors (**a**, top) by a blinded pathologist. Scale bar: 100 μm. **b** p-values of correlations between PBRM1 loss and p21 loss in human ccRCC samples. **c** A model depicting how PBRM1 may regulate tumor growth in ccRCC through recognition of p53 K382Ac. Source data are provided as a Source Data file.

Cancer Center IACUC committee (IRB#13–810). Forty cases from each of the four tumor stages (Stage I–IV) were randomly picked. Four foci from each tumor were used to generate TMA. Mice tumors were fixed with 200 ml 4% paraformaldehyde with stirring at room temperature overnight. Human ccRCC tumor TMA KD806 was purchased from US Biomax Inc.

Four micrometers of paraffin slides were deparaffinized in Shandon Varistain Gemini ES Autostainer. Antigen retrieval was performed with DAKO PTLink using Citrate Buffer (pH 6.0) at 98˚C for 20 min. Primary immunostaining was performed using antibodies against p21 (Cell-Signaling, cat#:#2947, 1:25), anti-p53 K382Ac antibody (GTX62061, 1:250), and PBRM1 (Bethyl labs, Cat# A301–591A, 1:50), Antibodies were incubated at room temperature for 30 min. Biotinylated anti-Rabbit (Vector Laboratories, cat#: BA-1000) secondary antibody and ABC-HRP complexes (Vector Laboratories, Cat#: PK6100) were applied following the primary antibodies. Each reagent was incubated for 30 min at room temperature. Three TBST washes were performed between steps. The signals were visualized with DAB substrate (DAKO, Cat#: K3468). Slides were then washed with DI water and processed with Hematoxylin counter stain, dehydrated and cleared in Shandon Varistain Gemini ES Autostainer. Finally the slides were coverslipped with Permount Mounting Medium.

Pathologist Dr. Wei Jiang performed the scoring of the stained foci. If greater than 50% of tumor cells were considered positive in a focus a score of 2 is given, 1 if <50% but greater than 5% of tumor cells were considered positive, and 0 if <5% of tumor cells were stained positive. If one marker is scored as 0 in one focus, then that whole tumor is deemed to have a score of 0 for that marker.

**Statistical analysis and reproducibility.** Individual in vitro experiments were performed three times unless otherwise indicated. The significance of three biological independent experimental data was determined by the two-tailed Student's paired t-test. P-value < 0.05 was considered significant and the numbers are indicated in the figures. Results are expressed as mean ± SEM. Categorical variables were compared between groups of TMA using Fisher's exact tests. No animals were excluded from experiments. No statistical method was used to predetermine sample size. Sample size was chosen on the basis of literature in the field.

**Reporting summary.** Further information on research design is available in the Nature Research Reporting Summary linked to this article.

## Data availability

Statistics source data for graphical representations and statistical analyses in Figs. 1–7, Supplementary Figs. 1–6 and Supplementary Table 1 are provided in PBRM1-p53-SourceData file. All the other data supporting the findings of this study are available from the corresponding author upon reasonable request.

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

## Acknowledgements

We are very grateful to Dr. Stephen Peiper for his support and guidance of this research. We thank Dr. Wei Xu at University of Wisconsin for providing the p-LNCX expression vector. We thank Dr. Hyemin Lee and Dr. Boyi Gan at MD Anderson Cancer Center for discussion. We thank Dr. Steven McMahon for discussions and suggestions. Research reported in this publication utilized the Translational Pathology Shared Resource at Sidney Kimmel Cancer Center at Jefferson Health and was supported by the National Cancer Institute of the National Institutes of Health under Award Number P30CA056036, a pilot award from VHL alliance (to H.F.Y.) and R01 CA211732 (to Q.Z.). The content is solely the responsibility of the authors and does not necessarily represent the official views of the NIH.

## Author contributions

W.J.C., L.Y.S., L. Li, W.J., and Z.J.Z. performed the experiments. W.J.C. wrote the first draft of the manuscript. L. La, Q.Y., Q.Z., and H.F.Y. assisted with editing. H.F.Y. made final decisions on the manuscript. H.F.Y. designed the experiments and oversaw the execution. E.D., J.R.T., and R.G.U. prepared and provided the TMA. W.J., Z.Z.L., and H.F.Y. analyzed the data. W.J.C., Z.Z.L., L. Li, Q.Z., and H.F.Y. prepared the figures. Q.Z. and H.F.Y. was responsible for funding acquisition.

## Competing interests

The authors declare no competing interests.
