## [Peer Review File · Nature Communications]

Reviewers' Comments:

Reviewer #1:

Remarks to the Author:

The manuscript by Cai et al. verifies PBRM1, a bromodomain-containing protein, as a reader of p53 CTD acetylation on lysine382 (K382Ac). PBRM1 has been previously described as a tumor suppressor in ccRCC, however the mechanism of PBRM1-mediated actions needs to be further elucidation. This manuscript suggests a correlation between the PBRM1 loss and the compromised p53 tumor suppressor function (p21 loss) in ccRCC tumors. The authors showed that 1) PBRM1 physically interacts with p53, which is strengthened upon DNA damage, 2) PBRM1 recognizes particular acetylation pattern containing K382Ac through its Bromodomain4, 3) Recognition of p53 K382Ac by PBRM1 is critical for p53's transcriptional activity on a subset of targets, such as CDKN1A (p21), 4) PBRM1 acts as a tumor suppressor in renal cancer by recognizing p53 K382Ac and regulating p53 signaling. This is an interesting study and most of the experiments are well-controlled and convincing. Nevertheless, there are a few concerns that need to be addressed.

1) The authors demonstrate that acetylation of K382 on p53 is critical for PBRM1 binding by pull-down assays using biotinylated p53 CTD peptides and co-IP assays using acetylation-loss p53-K382R mutant. It will be very interesting to see whether the acetylation-mimic p53-K382Q mutant could exhibit higher affinity with PBRM1 compared to WT p53.

2) In supplementary figure3a, the expression level of BD3 is very low compared to that of other BDs.

3) Figure.6a is not convincing. The expression levels of p21 and MDM2 only show very modest differences between PBRM1-wt and PBRM1-BD4* group. The PUMA expression shows significant difference only at 24h post-treatment. Another key point to notice is that the levels of PBRM1-wt and PBRM1-BD4* at 8h, 16h, 24h post-treatment are very inconsistent. More convincing data are needed to justify the main conclusion.

Reviewer #2:

Remarks to the Author:

In this manuscript, Cai et al found that, PBRM1 and p53 proteins interacted and mapped the interaction domain to the CTD of p53. The authors show that interaction is enhanced by p300 co-expression, which increases K382 acetylation. The investigators conducted in vitro experiments with acetylated peptides that showed increase binding with BD4 K382Ac. Reciprocally, deletion up to BD4 does not seem to impair binding and domains downstream of BD6 are dispensable. The authors then show that BD4/5 is sufficient to bind K382Ac. Mutations in BD4 abrogate increased binding to K382Ac leading the authors to conclude that BD4 is essential. The authors then found that KO PBRM1 reduces the ability of ectopic p53 to upregulate p21. Other p53 target genes behave similarly. The authors show that p53 and PBRM1 bind the p21 promoter. In a panel of kidney cancer cell lines they find that PBRM1 depletion downregulates p21 expression and that its overexpression induces p21. The authors go onto showing that this has the expected functional consequences in tumor growth using Ren xenografts and that there is a correlation between PBRM1 and p21 in ccRCC from patients by IHC.

Main

The notion that PBRM1 loss inactivates p53 does not fit with the observation that PBRM1-deficient tumors, unlike p53-deficient tumors, tend to be of low grade and aggressiveness.

The functional effects of PBRM1 in RCC should be examined in PBRM1-deficient RCC cells

reconstituted with PBRM1.

There is concern that the effects observed require DNA damaging agents. Are the authors able to observe K382Ac in RCC tumors?

Minor

The authors contend that mutation of BD4 abrogates K382Ac specific binding, but there is so much binding to non-Ac peptide that it is hard to draw conclusions.

The effects of p53 on p21 in Fig 4b are rather modest.

Dear Editors and reviewers:

We are very grateful to the comments and suggestions made by the editors and reviewers. We made our best effort to address them and our responses are listed below.

To address your comments, we have conducted experiments that are described in four reviewer figures below. The major experiments included in the revised manuscript are:

- (1) K382Q's impact on PBRM1 binding (reviewer Fig. 1);
- (2) The binding of individual BDs to PBRM1 when the expression levels are similar (reviewer Fig. 2, manuscript Fig. S3a);
- (3) Wild type PBRM1, but not BD4 mutant PBRM1, supports p53's ability to induce a subset of targets in PBRM1-deficient ccRCC cell lines after DNA damage (reviewer Fig. 3, manuscript Fig. 6a and b);
- (4) IHC analysis of K382Ac signal in human ccRCC (reviewer Fig. 4, manuscript Fig. S2d).

We also modified the text according to the reviewers' suggestions and critiques. We updated some references and acknowledgement. To facilitate your review of the revised manuscript, we marked the main change of the manuscript in **red**.

The reviewer comments (bold) and our responses:

Reviewer #1

The manuscript by Cai et al. verifies PBRM1, a bromodomain-containing protein, as a reader of p53 CTD acetylation on lysine382 (K382Ac). PBRM1 has been previously described as a tumor suppressor in ccRCC, however the mechanism of PBRM1-mediated actions needs to be further elucidation. This manuscript suggests a correlation between the PBRM1 loss and the compromised p53 tumor suppressor function (p21 loss) in ccRCC tumors. The authors showed that 1) PBRM1 physically interacts with p53, which is strengthened upon DNA damage, 2) PBRM1 recognizes particular acetylation pattern containing K382Ac through its Bromodomain4, 3) Recognition of p53 K382Ac by PBRM1 is critical for p53's transcriptional activity on a subset of targets, such as CDKN1A (p21), 4) PBRM1 acts as a tumor suppressor in renal cancer by recognizing p53 K382Ac and regulating p53 signaling. This is an interesting study and most of the experiments are well-controlled and convincing. Nevertheless, there are a few concerns that need to be addressed.

1) The authors demonstrate that acetylation of K382 on p53 is critical for PBRM1 binding by pull-down assays using biotinylated p53 CTD peptides and co-IP assays using acetylation-loss p53-K382R mutant. It will be very interesting to see whether the acetylation-mimic p53-K382Q mutant could exhibit higher affinity with PBRM1 compared to WT p53.

We thank the reviewer for this great suggestion. To address it, we generated p53-K382Q mutant and compared its affinity to PBRM1 with wild -type p53. We did not observe an increased affinity (reviewer Fig. 1a). To further confirm our observation, we synthesized a Biot-K382Q (368-393)

peptide to compare its affinity to PBRM1 with wild-type or Biot-K382Ac (369-393) peptides (reviewer Fig. 1b). Consistent with the previous result, the K382Q peptide behaved like the wild-type peptide. Previous reports also showed KQ mutant did not always mimic lysine acetylation well. (*A possible overestimation of the effect of acetylation on lysine residues in KQ mutant analysis*. J Comput Chem. 2012 Jan 30;33(3):239-46. *A KRAS GTPase K104Q Mutant Retains Downstream Signaling by Offsetting Defects in Regulation*. J Biol Chem. 2017 Mar 17;292(11):4446-4456). In addition, Sun et al. showed both KL and KQ mutants failed to mimic lysine acetylation in their system (*Acetylation of Beclin 1 inhibits autophagosome maturation and promotes tumour growth*. Nat Commun. 2015 May 26;6:7215.). Thus we conclude that although the K382Q mutation abolishes the positive charge of K382, it failed to fully recapitulate the characteristics of acetylated K382 on p53. A new comment was added on page 30 of the text.

Reviewer figure 1. **K382Q mutation on p53 does not increase its affinity to PBRM1.** A) Flag-PBRM1, wildtype or K382Q mutated Myc-p53 were transfected into H1299 cells with indicated combinations. Lysates were used for anti-Flag immunoprecipitation followed by Flag peptide elution. The inputs and eluates were immunoblotted with the indicated antibodies. B) H1299 cell lysates were incubated with indicated biotinylated p53 peptides. The peptides were pulled down with streptavidin beads and the associated protein was immunoblotted with indicated antibodies.

2) In supplementary figure 3a, the expression level of BD3 is very low compared to that of other BDs.

New experiment was performed with similar expression levels of BDs (reviewer Fig. 2, manuscript Fig. S3a). The conclusion is the same and the old figure was replaced in the manuscript.

Reviewer figure 2, manuscript supplemental figure S3a. HCT116 cells were transfected with vector or individual Flag-PBRM1 bromodomains and treated with 50 μ M etoposide for 8 h. Lysates were subjected to immunoprecipitation with Flag-M2 beads. The inputs and eluates were analyzed by immunoblots.

3) Figure.6a is not convincing. The expression levels of p21 and MDM2 only show very modest differences between PBRM1-wt and PBRM1-BD4* group. The PUMA expression shows significant difference only at 24h post-treatment. Another key point to notice is that the levels of PBRM1-wt and PBRM1-BD4* at 8h, 16h, 24h post-treatment are very inconsistent. More convincing data are needed to justify the main conclusion.

We agree that this is a key experiment that tests our hypothesis. We repeated the experiment with Ren-01 cells with PBRM1 KO expressing various constructs. In this set of experiment, difference in p21 was more pronounced at 16h and 24h post treatment, while PUMA difference was more obvious after 8h or 24h of treatment. MDM2 difference was obvious at 8h and 16h after treatment. Moreover, the PBRM1-wt and PBRM1-BD4* expression levels were comparable at various time points (reviewer Fig. 3, manuscript Fig. 6a).

To more rigorously test our hypothesis, we expressed GFP, PBRM1-wt or PBRM1-BD4* in a PBRM1-null ccRCC cell line SLR24. The cells were treated the same way as in Fig. 6a and the induction of p53 downstream targets was compared. In SLR24 cells, p21 or PUMA were barely induced by etoposide treatment. When PBRM1-wt is expressed, both were strongly induced at all the time points, but PBRM1-BD4*, although expressed at the same levels as the wild type protein, failed to induce their expression. The difference of MDM2 induction was also obvious at 8h post treatment (reviewer Fig. 3, manuscript Fig. 6b). Thus in various PBRM1-null ccRCC cell lines, our data strongly suggests that wild type but not BD4 mutant PBRM1 assists p53 to induce a subset of its downstream targets. The changes were made in pages 23-25 in the text.

Reviewer figure 3, manuscript supplemental figure 6 ab. **Wild type PBRM1, but not BD4 mutant PBRM1, supports p53's ability to induce a subset of targets in PBRM1-deficient ccRCC cell lines after DNA damage.** a. GFP, wild-type or BD4* mutant PBRM1 were stably expressed in Ren-01 PBRM1 KO cells (combination of three clones) a) or PBRM1-null SLR24 cells b). Cells were treated with 50 μ M etoposide for the indicated times, and lysates were analyzed via immunoblots with indicated antibodies. The band intensity of indicated protein was measured with Bio-rad Image Lab 4.1, and the relative ratios were calculated over the signal intensity of Vinculin in the corresponding lanes.

Reviewer #2.

In this manuscript, Cai et al found that, PBRM1 and p53 proteins interacted and mapped the interaction domain to the CTD of p53. The authors show that interaction is enhanced by p300 co-expression, which increases K382 acetylation. The investigators conducted in vitro experiments with acetylated peptides that showed increase binding with BD4 K382Ac. Reciprocally, deletion up to BD4 does not seem to impair binding and domains downstream of BD6 are dispensable. The authors then show that BD4/5 is sufficient to bind K382Ac. Mutations in BD4 abrogate increased binding to K382Ac leading the authors to conclude that BD4 is essential. The authors then found that KO PBRM1 reduces the ability of ectopic p53 to upregulate p21. Other p53 target genes behave similarly. The authors show that p53 and PBRM1 bind the p21 promoter. In a panel of kidney cancer cell lines they find that PBRM1 depletion downregulates p21 expression and that its overexpression induces p21. The authors go onto showing that this has the expected functional consequences in tumor growth using Ren xenografts and that there is a correlation between PBRM1 and p21 in ccRCC from patients by IHC.

Main

4) The notion that PBRM1 loss inactivates p53 does not fit with the observation that PBRM1-deficient tumors, unlike p53-deficient tumors, tend to be of low grade and aggressiveness.

We thank the reviewer for this thoughtful comment. We think this discrepancy can be explained by the fact that PBRM1 deficiency only compromise, not abolish, p53 function on a subset of p53 targets, thus it is unlikely to recapitulate all the attributes of p53 deficiency. A new comment on this was added to page 33.

5) The functional effects of PBRM1 in RCC should be examined in PBRM1-deficient RCC cells reconstituted with PBRM1.

This was done in manuscript Fig. 5e (RCC4) and Fig. 6b (SLR24) (see above). All results are consistent with our hypothesis.

6) There is concern that the effects observed require DNA damaging agents. Are the authors able to observe K382Ac in RCC tumors?

Yes. We were able to detect K382Ac IHC signals in many samples on a tissue microarray of ccRCC. The signals are located in the nucleus of the cancer cells within the tumors (reviewer Fig. 4, manuscript Fig. S2b). Since DNA damaging agents are not a part of standard of care for ccRCC patients, this result suggests that K382Ac occurs naturally without treatment of DNA damaging agents. The change is made on page 7-8 in the text.

Reviewer figure 4, manuscript figure s2b. **K382Ac IHC signals are detected in human ccRCC tumor.** 200x IHC images of representative ccRCC (top) focus stained with anti-K382Ac antibody. A kidney cancer tissue array KD806 from US Biomax Inc. was used for this analysis.

Minor

7) The authors contend that mutation of BD4 abrogates K382Ac specific binding, but there is so much binding to non-Ac peptide that it is hard to draw conclusions.

We apologize for not making our point clear. We believe that the non-acetylated p53 peptide already binds PBRM1, and K382Ac enhances this interaction. Mutation of BD4 abrogates this enhanced binding. The text was changed on page 28.

8) The effects of p53 on p21 in Fig 4b are rather modest.

This was likely caused by the over-exposure of p21. A weaker exposure is now used which shows that the difference is significant.

Reviewer figure 5, manuscript figure 4b. **PBRM1 is required for full p53 transcriptional activity on a subset of its targets.**

b. H1299 parental and PBRM1-KO#1 cells were transfected with increasing amounts of p53 and lysates were immunoblotted.

Thank you for the consideration on behalf of all the co-authors!

Haifeng

Haifeng Yang, Ph.D.
Assistant Professor
Department of Pathology, Anatomy and Cell Biology
Thomas Jefferson University
JAH336D

1020 Locust Street
Philadelphia, PA 19107
Office: 215-503-6163

Reviewers' Comments:

Reviewer #1:

Remarks to the Author:

All the issues have been well addressed. The revised manuscript is acceptable for publication.

Reviewer #2:

None